# Interventions to Improve Vaccination Uptake Among Adults: A Systematic Review and Meta-Analysis

**DOI:** 10.3390/vaccines13080811

**Published:** 2025-07-30

**Authors:** Anelisa Jaca, Lindi Mathebula, Thobile Malinga, Kimona Rampersadh, Masibulele Zulu, Ameer Steven-Jorg Hohlfeld, Charles Shey Wiysonge, Julie C. Jacobson Vann, Duduzile Ndwandwe

**Affiliations:** 1Cochrane South Africa, South African Medical Research Council, Cape Town 7505, South Africa; lindi.mathebula@mrc.ac.za (L.M.); thobile.malinga@mrc.ac.za (T.M.); kimona.rampersadh@mrc.ac.za (K.R.); masibuzulu@gmail.com (M.Z.); ameer.hohlfeld@mrc.ac.za (A.S.-J.H.); duduzile.ndwandwe@mrc.ac.za (D.N.); 2Vaccine Preventable Diseases Programme, World Health Organization Regional Office for Africa, Brazzaville P.O. Box 06, Congo; wiysonge@yahoo.com; 3School of Nursing, The University of North Carolina at Chapel Hill, Chapel Hill, NC 27599, USA; jvann@email.unc.edu

**Keywords:** vaccination uptake, adult population, effective interventions, meta-analysis

## Abstract

Background: Immunization is a highly effective intervention for controlling over 20 life-threatening infectious diseases, significantly reducing both morbidity and mortality rates. One notable achievement in vaccination efforts was the global eradication of smallpox, which the World Health Assembly declared on 8 May 1980. Additionally, there has been a remarkable 99.9% reduction in wild poliovirus cases since 1988, decreasing from more than 350,000 cases that year to just 30 cases in 2022. Objectives: The objective of this review was to assess the effects of various interventions designed to increase vaccination uptake among adults. Search Methods: A thorough search was conducted in the CENTRAL, Embase Ovid, Medline Ovid, PubMed, Web of Science, and Global Index Medicus databases for primary studies. This search was conducted in August 2021 and updated in November 2024. Selection Criteria: Randomized trials were eligible for inclusion in this review, regardless of publication status or language. Data Analysis: Two authors independently screened the search outputs to select potentially eligible studies. Risk ratios (RR) with 95% confidence intervals (CI) were calculated for each randomized controlled trial (RCT). A meta-analysis was conducted using a random-effects model, and the quality of the evidence was assessed using the GRADE approach. Main Results: A total of 35 randomized controlled trials met the inclusion criteria and were included in this review, with the majority conducted in the United States. The interventions targeted adults aged 18 and older who were eligible for vaccination, involving a total of 403,709 participants. The overall pooled results for interventions aimed at increasing influenza vaccination showed a risk ratio of 1.41 (95% CI: 1.15, 1.73). Most studies focused on influenza vaccination (18 studies), while the remaining studies examined various other vaccines, including those for hepatitis A, COVID-19, hepatitis B, pneumococcal disease, tetanus, diphtheria, pertussis (Tdap), herpes zoster, and human papillomavirus (HPV). The results indicate that letter reminders were slightly effective in increasing influenza vaccination uptake compared to the control group (RR: 1.75, 95% CI: 0.97, 1.16; 6 studies; 161,495 participants; low-certainty evidence). Additionally, participants who received education interventions showed increased levels of influenza vaccination uptake compared to those in the control group (RR: 1.88, 95% CI: 0.61, 5.76; 3 studies; 1318 participants; low-certainty evidence). Furthermore, tracking and outreach interventions also led to an increase in influenza vaccination uptake (RR: 1.87, 95% CI: 0.78, 4.46; 2 studies; 33,752 participants; low-certainty evidence). Conclusions: Letter reminders and educational interventions targeted at recipients are effective in increasing vaccination uptake compared to control groups.

## 1. Introduction

Globally, vaccine-preventable diseases (VPDs) cause millions of deaths each year [1]. Vaccination involves administering a vaccine, either orally or through injection, to stimulate an immune response against a specific disease [2]. Immunization ensures that vaccinated individuals are protected from that disease. Vaccination is an evidence-based intervention for controlling more than 20 life-threatening infectious diseases, significantly reducing both morbidity and mortality [3]. One of the most notable achievements in vaccination history is the worldwide eradication of smallpox, officially declared by the World Health Assembly on 8 May 1980 [1]. Another significant success is the reduction in wild poliovirus cases by 99.9% since 1988, decreasing from more than 350,000 cases to just 175 in 2019 [4,5].

Type 2 wild poliovirus was declared eliminated in 2015, followed by type 3, in 2018. Five regions of the World Health Organization (WHO) successfully eliminated wild poliovirus, apart from the Eastern Mediterranean Region, where polio remains endemic in Afghanistan and Pakistan [6]. The WHO estimates that vaccination prevents between four million and five million deaths annually [3]. Additionally, the Vaccine Impact Modeling Consortium, comprising 16 independent research groups, conducted modeling to estimate the impact of vaccination programs on disability-adjusted life years and deaths for ten pathogens across 98 low- and middle-income countries. It is estimated that between 2000 and 2030, vaccination will prevent 69 million deaths in these countries [7]. In the United States, during the 2019–2020 influenza season, influenza vaccination is estimated to have averted approximately 7.5 million cases, 105,000 hospitalizations, 3.69 million medical visits, and 6300 deaths [8].

### 1.1. Recommended Vaccines

Vaccination recommendations for adults differ significantly across countries in terms of the number of vaccines, types of vaccinations, target populations, and whether they are recommended or required [8,9,10]. The requirements vary based on factors, such as age, existing health conditions, and vaccination history [10,11]. According to the Centers for Disease Control and Prevention (CDC), the recommended vaccines for adults include those against COVID-19, influenza, and either the Tetanus, Diphtheria, and Pertussis (Tdap) or the Tetanus and Diphtheria (Td) [12]. For pregnant women, the recommended vaccinations are the Tdap vaccine, which should be administered between 27 and 36 weeks of pregnancy, along with COVID-19, influenza, and hepatitis B vaccines. Healthcare workers (HCWs) are often prioritized for vaccines against COVID-19, Chickenpox, influenza, hepatitis B, Meningococcal, Measles, Mumps, Rubella (MMR), and Tdap or Td. This is due to their increased risk of exposure to life-threatening infections when they are involved in direct patient care [13,14]. International travelers, immigrants, and refugees are advised to be fully vaccinated against MMR and hepatitis A. Additionally, there are specific vaccination recommendations for individuals with certain health conditions. For instance, individuals with asplenia should receive Hib, Meningococcal, and pneumococcal vaccines. Individuals with type 1 or type 2 diabetes should receive the pneumococcal vaccine, and those with heart disease, stroke, or other cardiovascular diseases are also recommended to receive the pneumococcal vaccine. People living with HIV are advised to receive hepatitis A, hepatitis B, Meningococcal conjugate, pneumococcal, Shingles, Chickenpox, and MMR vaccines. People with liver disease should be vaccinated against hepatitis A, hepatitis B, and pneumococcal diseases. Finally, those with lung diseases, namely, asthma and chronic obstructive pulmonary disease, as well as individuals with end-stage renal disease, are recommended to receive hepatitis B and pneumococcal vaccines [12]. There is a lack of data regarding adult vaccination recommendations in low- and middle-income countries.

### 1.2. Vaccination Uptake Among Adults

Vaccination offers significant health benefits to communities; however, adult vaccination uptake remains low globally, leaving many adults unprotected against vaccine-preventable diseases [11,15]. It is important to note that vaccination coverage also varies by country. In 2022, the United States reported the following vaccination coverage among adults for different vaccines: pneumococcal vaccination for individuals aged 19 to 64 years was at 23.0%, herpes zoster vaccination for those aged 19 years and older was 34.4%, Tdap and Td vaccinations for adults aged 19 years and older was 59.4%, HPV vaccination for females aged 19 to 26 years was 57.7%, and COVID-19 vaccination coverage among adults aged 19 years and older was 79.7% [12]. In Australia, vaccination coverage in 2023 showed that 41.0% of adults aged 71 years received the herpes zoster vaccine, and 41.7% of adults aged 72 years received the pneumococcal vaccine [16]. A population-based study conducted in India between 2017 and 2018 reported a 2% vaccination coverage among adults aged 45 years and older for influenza, pneumococcal, typhoid, and hepatitis B vaccines. In Canada, vaccination coverage was reported in 2023 for measles (87%), HPV (18%), pneumococcal vaccination among adults aged 65 years and older (55%), influenza vaccination among adults aged 65 years and older (70.2%), and one dose of COVID-19 vaccine (94.5%). Additionally, vaccination coverage among travelers was reported as follows: COVID-19 (81.4%), hepatitis A (16.7%), yellow fever (7.7%), cholera and traveler’s diarrhea (7.6%), and typhoid vaccines (6.2%) [17].

Significant disparities exist in COVID-19 vaccination rates across different countries [1,18] As of 16 July 2021, a total of 3.6 billion doses of COVID-19 vaccines had been administered globally, averaging about 30 million doses per day. While more than 25% of the world’s population had received at least one dose of a COVID-19 vaccine, only 1% of individuals in low-income countries had received at least one dose [19]. In high-income countries, e.g., the UK and Israel, approximately 60% of adults had been vaccinated with at least one dose. In contrast, vaccination coverage in low- and middle-income countries was as follows: 30% in Morocco, 26% in Colombia, 23% in India, 22% in Russia, 7% in South Africa, and less than 1% in many countries in sub-Saharan Africa [19]. The low vaccination rates in these regions have resulted in a significant burden of morbidity and mortality from diseases that could have been prevented by vaccination.

### 1.3. Interventions to Improve Vaccine Uptake Among Adults

To improve vaccination uptake, several interventions can be implemented. These interventions can target vaccination recipients, providers, and health systems [20,21,22,23,24,25]. The interventions may also focus on structural changes, broader system improvements, or policy modifications [18,26]. Person-focused interventions include educating individuals on the importance of asking healthcare providers to check their vaccination records and reminding them to stay up to date with their vaccinations. Health education programs and communication strategies—such as presumptive communication, storytelling, motivational interviewing, and applying the Health Belief Model to design messages—can also be effective in increasing vaccination rates. Furthermore, direct advertising and promotional campaigns from pharmaceutical companies, health insurance providers, and health systems can help raise awareness. Lastly, offering incentives can motivate individuals to get vaccinated [21,23,27].

Provider-focused interventions involve various strategies, namely, training, supportive supervision, educational outreach visits, provider reminders, clinical decision support tools, audit and feedback, academic detailing, provider incentives, standing vaccination orders, and the appointment of a vaccination champion within clinical practices [20,23,27,28] Systems-level interventions involve modifying practices at healthcare clinics, implementing population health strategies, utilizing electronic health records, systematically screening vaccination histories of individuals admitted to hospitals, making vaccination services more accessible by bringing them closer to consultation rooms, and providing vaccinations in non-traditional locations such as inpatient units, emergency departments, businesses, community-based settings, and during home visits. Expanding clinic hours to accommodate patient schedules is also important [20,25,27]. Broader systems- or policy-level interventions include mandates for vaccination as a requirement for employment, school, or college matriculation, or a combination of these; the establishment of vaccination registries; offering free vaccines or reducing out-of-pocket costs for patients; and ensuring health insurance coverage for vaccinations [18].

These interventions may be effective by addressing barriers to access and increasing demand for vaccinations [29,30,31]. Factors associated with low vaccination uptake among adults can be categorized into patient-level, provider-level, health system-level, and policy-level barriers [18,32]. At the patient level, barriers include lack of access, lack of health insurance coverage, affordability, concerns about safety and potential adverse effects, insufficient awareness or knowledge, low health literacy, the belief that healthy individuals do not need vaccinations, mistrust of pharmaceutical companies, forgetting vaccination appointments, not receiving a clear recommendation from healthcare providers, not having routine visits with primary care providers, and fear of needles [18,32,33].

Provider-level barriers include forgetting to offer vaccinations, financial costs to the practice, concerns about reimbursement and costs to patients, storage challenges, knowledge gaps, attitudes, and beliefs regarding safety and efficacy, time constraints, lack of confidence or discomfort when discussing vaccines, not being up-to-date with their vaccinations, and lack of trust in their institutions [34,35] Health system barriers encompass insufficient infrastructure for delivering adult vaccinations, viz., inadequate funding, supply chain or distribution challenges, absence of population management systems, and a lack of provider reminder systems, clinical decision support tools, and patient reminder or recall systems [18,21,33,36].

Adult vaccination is crucial because adults are vulnerable to infectious diseases, especially as their immune systems may become compromised with age. Older adults face an increased risk of severe outcomes from infectious diseases due to the natural aging of their immune systems [37]. Changes in chemokine localization and lymph node structure can affect immune function in older individuals [38,39]. Emerging infections, such as Ebola and COVID-19, also pose significant threats, making both younger and older adults primary targets for vaccination. For these reasons, the WHO, the U.S. CDC, and the European CDC recommend vaccinations for adults.

Despite the importance of vaccinations, uptake among adults remains low, falling short of achieving vaccination targets. This low uptake can be traced back to barriers at the patient, provider, health system, and policy levels [18]. Implementing effective strategies to enhance vaccination rates among adults can lead to reduced morbidity, mortality, and healthcare costs [40]. Consequently, more widespread interventions are needed to address existing barriers and improve vaccination uptake among adults.

### 1.4. Objectives

The objective is to evaluate the effects of interventions aimed at increasing vaccination uptake among adults.

## 2. Methods

### 2.1. Protocol Registration

The protocol for this systematic review was not registered in any database.

### 2.2. Criteria for Considering Studies for This Review

We included randomized controlled trials (RCTs) regardless of their publication status or language. Eligible participants included adults who were eligible for vaccines approved for use in their respective countries. This group encompassed caregivers of older adults and all types of healthcare workers (HCWs) targeted for vaccinations. We included studies that reported outcomes specifically about this population. Our focus was on interventions designed to enhance the uptake of routine vaccines, including those for specific medical conditions or special indications. Eligible interventions targeted vaccination recipients, vaccination service providers, and health systems, including policy changes. We excluded trials that assessed interventions aimed at improving the uptake of travel vaccines, as these are often mandatory for travelers.

### 2.3. Selection of Studies

Retrieved articles were uploaded to Covidence, where duplicates were removed. The titles and abstracts were screened by two authors (AJ and TM or LM and MS), independently. The same authors also screened the full-text articles for inclusion. Any disagreements that arose during this process were resolved by consulting other review authors (CSW, DN, and JJV). A “Characteristics of Excluded Studies” table is included for studies that did not meet our eligibility criteria (Appendix A).

Data Extraction and Management.

We utilized the Effective Practice and Organisation of Care (EPOC) standard data collection form, which had been tested in one of the included studies, to extract data [41]. Data were independently extracted by AJ and TM or LM and MS, focusing on the following study characteristics:

Methods: study design, setting, study duration, and follow-up times.

Participants: number, age, and sex.

Interventions: type of intervention and comparator.

Outcomes: primary and secondary outcomes.

### 2.4. Data Synthesis and Statistical Analysis

We conducted pooled meta-analyses when the interventions, participants, and vaccines were similar [42]. All statistical analyses were performed using R (version 12.1). The metapackage was used for conducting meta-analyses of binary outcomes, calculating risk ratios (RRs) and 95% confidence intervals (CIs) using the Mantel–Haenszel method for pooling, along with the DerSimonian–Laird estimator to account for random effects. The dataset was initially stratified by predefined analysis groups, with each subgroup analyzed separately. For each analysis, we used the ‘metabin()’ function to synthesize data on intervention and control events, along with total sample sizes, organized by study and further stratified by subgroup where applicable. Forest plots were generated for each analysis using the meta and grid packages, ensuring customized formatting for clarity and consistent visual presentation.

## 3. Results

### 3.1. Description of Studies

#### 3.1.1. Results of the Search

Our literature search yielded a total of 12,754 records from electronic databases and other sources. We identified and excluded 20 duplicate records from these datasets. Consequently, we screened 12,734 studies, out of which 12,082 were deemed ineligible for inclusion in our review. We assessed the full texts of the remaining 652 potentially eligible studies and excluded 617 of them, providing reasons for exclusion (Appendix A).

Ultimately, 35 RCTs were deemed eligible for inclusion. The PRISMA diagram illustrates the study selection process (Appendix A).

#### 3.1.2. Included Studies

The RCTs included individuals as the unit of randomization. The duration of these RCTs ranged from 2 months to 3 years and 4 months (Appendix A). Most of the included RCTs were conducted in the United States [43,44,45,46,47,48,49,50,51,52,53,54,55,56,57,58,59,60,61,62,63,64,65,66], while others were carried out in Australia [67,68,69,70], Belgium [71], Switzerland [72], Hong Kong [73], Denmark [74], England, and Hawaii [75,76].

##### Participants

All included studies examined the effects of interventions aimed at increasing vaccination uptake among adults aged 18 years and older who were eligible for vaccination, totaling 403,709 participants. Most studies further specified the type of adults investigated, including those at high risk of influenza complications [76], individuals who underwent a health check between the end of the 2016 influenza epidemic and the beginning of the next vaccination campaign [72], active patients at participating primary care centers [52], and those deficient in influenza, Tdap, or pneumococcal vaccines [53]. Participants also included adults with chronic diseases, individuals with vaccine indications who had not been previously vaccinated [54], those who had never been tested for hepatitis B [56], attending healthcare services during the study duration (study 69), and who were either negative or positive for hepatitis C virus (HCV) or had unknown status [55]. Other categories included adults who had not received the herpes zoster vaccine [63,65], individuals starting their first HPV vaccine [73], direct healthcare providers [47,50,74,75], and women who had their first trimester obstetric visit during the study period [45]. In addition, adults missing pneumococcal and herpes zoster (HZV) vaccinations, those aged 18 to 64 with no diabetes, adults aged 65 and older without diabetes, and those aged 18 and older with diabetes [48] were also included in the studies.

##### Interventions and Comparisons

A summary and detailed description of the interventions and comparisons used in the included studies are presented in the Characteristics of Included Studies table. The interventions are stipulated as recipient- and provider-oriented interventions.

##### Outcomes

Most included studies reported data on our primary outcome, i.e., vaccination uptake. In total, 18 studies evaluated the effects of various recipient and provider interventions on influenza vaccination uptake [44,45,47,48,50,51,52,53,59,61,62,67,72,73,74,75,76]. Some of the studies also investigated the effects of the interventions on uptake of other vaccines, i.e., COVID-19, hepatitis A virus (HAV), hepatitis B virus (HBV), pneumococcal, tetanus, Tdap, HZV, and HPV [43,49,53,54,55,56,62,63,66,68,69,70,71]. The secondary outcomes, proportion of people who are ‘up-to-date’ with needed vaccinations, and other vaccination indicators were not reported in the included studies.

### 3.2. Effects of Interventions

#### 3.2.1. Recipient-Oriented Interventions

##### Overall Reminders Versus Control or Standard Care

The overall pooled result for interventions increasing influenza vaccination was RR: 1.41 with 95% CI: 1.15, 1.73. For other vaccinations, the overall pooled results were RR: 2.69, with a 95% CI: 1.79, 4.04.

Patients receiving reminder interventions were more likely to have received the influenza vaccine compared to those in the control group (RR: 1.28, 95% CI: 0.85, 1.93). Similarly, patients in the intervention group were more likely to have received other vaccinations, i.e., HBV, pneumococcal, and herpes zoster vaccines (RR: 1.46, 95% CI: 1.05, 2.03) compared to those in the control.

##### Different Types of Reminders on Vaccination Uptake

We assessed the effects of different types of reminders on vaccination uptake; our results indicate that letter reminders were quite effective in increasing influenza vaccination uptake compared to the control (RR: 1.75, 95% CI: 0.97, 3.17). There was a slight increase in influenza vaccination uptake among participants who received a postcard (RR: 1.11, 95% CI: 1.05, 1.16), and phone call (RR: 1.19, 95% CI: 1.12, 1.26), while there was no difference among those who received message reminders (RR: 1.00, 95% CI: 0.89, 1.14). For participants who received reminders, there was an increase in uptake of HBV, pneumococcal, and herpes zoster vaccines in letter reminders (RR: 1.68, 95% CI: 1.04, 2.70) compared to phone call reminders (RR: 1.32, 95% CI: 0.88, 1.98) (Appendix A).

##### Education Versus Control or Standard Care

There was an increase in influenza vaccination uptake among participants who received education interventions compared to those in the control group (RR: 1.88, 95% CI: 0.61, 5.76). Compared to the control group, participants in the intervention were also more likely to receive other vaccinations, like the HAV, HBV, pneumococcal, Tdap, and herpes zoster vaccines (RR: 4.30, 95% CI: 2.96, 6.25) (Appendix A).

##### Tracking and Outreach Versus Control or Standard Care

There was an increased likelihood of influenza vaccination uptake among participants in the tracking and outreach group compared to participants in the control group (RR: 1.87, 95% CI: 0.78, 4.46) (Appendix A).

##### Promotional Campaigns vs. Control or Standard Care

Promotional campaigns showed small differences in influenza vaccination uptake among participants in the promotional campaign group compared to those in standard care (RR: 1.07, 95% CI: 1.02, 1.13, Appendix A).

##### Financial Incentives

The associations between financial incentives and hepatitis B vaccination completion (RR: 1.31, 95% CI: 1.07, 1.59) and COVID-19 (second dose) vaccination uptake (RR: 1.22, 95% CI: 0.85, 1.74) were relatively small, but important (Appendix A).

#### 3.2.2. Provider-Oriented Interventions

##### Letter Reminders Versus Control or Standard Care

In the primary care centers, providers in the letter reminder group had an increased influenza vaccination uptake compared to those in the control group (RR: 1.76, 95% CI: 0.65, 4.75) (Appendix A).

##### Risk of Bias in Included Studies

Allocation random sequence generation had a low risk of bias in 19 studies [42,44,47,50,51,52,56,57,58,61,62,63,64,65,68,69,72,73], unclear risk in 9 studies [43,46,48,49,53,54,60,67,72] and high risk in 4 studies [49,56,57,74] Allocation concealment had a low risk of bias in 10 studies [44,50,51,52,57,58,59,61,62,73], unclear risk in 15 studies [43,47,48,49,53,54,56,60,62,63,67,68,69,71,72] and high risk in 5 studies [55,64,66,70]

##### Blinding

Regarding blinding of participants and personnel, there was a low risk of bias in 11 studies [44,48,50,52,57,61,62,63,64,69,73], unclear risk in 16 studies [42,43,47,49,51,53,54,55,56,58,59,60,67,68,70,72] and high risk in 4 studies [45,65,66,71]. Blinding of outcome assessment had a low risk of bias in 11 studies [43,44,52,53,56,57,61,62,63,64,67], unclear risk in 13 studies [47,48,49,50,51,54,55,60,68,69,70,71,72], and high risk in 2 studies [65,66]

##### Incomplete Outcome Data

There was a low risk of bias in 23 studies [45,48,49,51,52,53,55,56,57,58,59,60,61,62,63,64,65,66,67,68,70,71,73] unclear risk in 3 studies [43,44,77] and high risk in 4 studies [47,50,54,72] under incomplete outcome data.

##### Selective Reporting

A total of 25 studies were rated as having a low risk of bias [44,45,47,49,50,51,52,53,56,57,58,59,60,61,62,63,64,66,67,68,70,71,72,74,77], 4 studies were considered to have an unclear risk of bias [43,48,54,55] and 2 as having a high risk of bias [43,44].

##### Other Potential Sources of Bias

A total of 20 studies had no evidence of other sources of bias [43,44,45,47,51,52,53,54,57,59,60,61,63,65,68,71,72,73,74,77] while 5 had unclear risk [48,49,50,55,64], and 4, a high risk [62,66,67,70]. Other types of potential risk of bias included missing data, funder and conflicts of interest statement not mentioned, recall bias by the participants, providers not having checked the patient’s vaccination record since they assumed that the absence of a reminder indicated prior vaccination, and some of the patients in the control group knowing the people who had received an invitation for free influenza vaccination.

##### Excluded Studies

Our reasons for excluding some studies from this review are briefly specified in the Characteristics of Excluded Studies table (Appendix A). The predominant reasons for exclusion were irrelevant study designs, that is, cluster RCTs, reviews, or ongoing clinical trials. Other reasons included unrelated comparison groups, where some of the studies compared two interventions without a control group. Some assessed the effects of interventions on outcomes that were not related to our review question.

#### 3.2.3. Sensitivity Analysis

Our sensitivity analysis, after excluding each study, showed consistent results, as the overall effect size (RR: 1.61) remained unchanged (Appendix A).

## 4. Discussion

### 4.1. Summary of Main Findings

This systematic review examined the effectiveness of recipient-oriented interventions, including reminders (letters, postcards, phone calls, and text messages), educational strategies, financial incentives, tracking and outreach, and promotional campaigns, on adult vaccination uptake and completion. It also evaluated provider-oriented interventions, i.e., letter reminders and educational interventions aimed at increasing vaccination among eligible patients. Among the recipient-directed interventions, letter reminders showed the strongest association with increased influenza vaccination uptake, although the effects were modest and based on low-certainty evidence. Smaller effects were observed for postcard and phone call reminders, while message reminders showed no impact. Some evidence suggested that letter reminders might be linked to improved uptake of other vaccines, such as HBV (hepatitis B vaccine), pneumococcal, and HZV (herpes zoster vaccine). Educational interventions, tracking and outreach activities, and, to a lesser extent, financial incentives were also associated with increased vaccination in some studies. However, most findings were derived from studies assessed as having low certainty due to methodological limitations. In terms of provider-oriented interventions, some studies indicated that letter reminders and educational strategies directed at providers could contribute to modest improvements in patient vaccination uptake, particularly for influenza. Nonetheless, these results should be considered carefully, as the evidence base is limited and of low certainty.

### 4.2. Overall Completeness and Applicability of the Evidence

This review synthesizes evidence from 35 studies evaluating interventions aimed at increasing adult vaccination uptake. The majority of these studies were conducted in high-income countries (HICs), primarily in the United States, Canada, and European nations. As a result, the applicability of the findings to low- and middle-income countries (LMICs) remains uncertain. Health system infrastructure, population health literacy, communication channels, and access to care vary significantly across different settings, and these contextual factors can influence the effectiveness of an intervention. For example, mailed letter reminders may work well in areas with reliable postal systems but may not be feasible or impactful in regions lacking such infrastructure. Moreover, few studies explicitly addressed how sociocultural, economic, or systemic barriers, such as vaccine misinformation, distrust in healthcare systems, or geographic inequities in service access, might affect the uptake of these interventions. Therefore, the transferability of interventions, especially those requiring substantial resources or provider training, should be approached with caution. Given these contextual differences, policymakers and implementers must assess local needs and constraints before adapting or scaling up any interventions. Further research is needed to evaluate the effectiveness of these strategies in diverse and under-represented settings.

### 4.3. Certainty of the Evidence

Of the included RCTs, 25 were judged to be at high risk of bias, 3 at unclear risk, and only 7 at low risk of bias. Common methodological concerns included selective reporting, lack of allocation concealment, and absence of blinding. Consequently, the overall certainty of the evidence was rated low for most interventions. While the findings indicate possible beneficial effects, they should be interpreted with caution. Future high-quality studies are needed to better understand the true impact of these interventions.

### 4.4. Potential Biases in the Review Process

Efforts were made to minimize bias by adhering to Cochrane guidance. However, the possibility of language bias exists, as only English-language studies were included, potentially resulting in the omission of relevant studies published in other languages.

### 4.5. Comparison with Other Reviews

Few systematic reviews have focused on interventions to enhance adult vaccination uptake, and are among the most relevant [21,77,78]. Similarly to our review, they found that reminder interventions, including letters, phone calls, and educational efforts, were associated with increased influenza vaccination. However, unlike the current review, which identified a predominance of high-risk-of-bias studies, earlier reviews reported studies of mostly moderate risk. Jacobson Vann also assessed thse effectiveness of text messages and combinations of patient and provider reminders. The study by [77] concentrated on interventions to increase COVID-19 vaccination, reporting positive effects from financial incentives and targeted communication strategies. Although the majority of our studies did not focus on COVID-19 vaccination, the overall pattern of findings aligns with previous work that suggests multi-component and personalized interventions may offer the most promise.

### 4.6. Implementation Considerations, Cost, and Equity

While this review focused on the effectiveness of various interventions, the included studies provided limited information on implementation challenges, cost-effectiveness, and equity-related outcomes. These factors are critical for decision-makers, particularly in settings with limited health budgets and resources. Interventions such as phone reminders, provider education, or tracking systems may require staffing, technological infrastructure, or financial investments that are not available in all settings. For instance, digital interventions may be effective in urban populations with high digital literacy but less so in rural or underserved communities. Additionally, few studies reported cost analyses or considered the implications of resource allocation, making it difficult to assess the feasibility of scaling up these interventions in real-world public health systems. Equity was also insufficiently addressed, with only a small number of studies explicitly targeting or reporting outcomes among populations at higher risk of under-vaccination, such as minority groups, individuals with lower socioeconomic status, or those with limited access to healthcare. This gap represents a missed opportunity, as interventions that improve equity in vaccination uptake are essential for achieving broader public health goals. Looking ahead, future studies and implementation efforts should consider incorporating elements, namely, equity-focused design (e.g., targeting underserved populations), cost-effectiveness analyses, and process evaluations that assess feasibility, acceptability, and sustainability in different health system contexts.

## 5. Conclusions

### 5.1. Implications for Practice

Our review suggests that recipient-oriented interventions, i.e., letter reminders, education sessions, and tracking and outreach, may be associated with improved vaccination uptake. Postcards and phone interventions showed small potential effects, while message reminders were generally not associated with changes in uptake. Provider-directed letter reminders may also be linked to modest improvements. However, the overall certainty of the evidence was low, limiting the strength of these conclusions. Therefore, when considering the use of these interventions, implementation scientists, clinicians, public health agencies, and health systems should proceed with caution and prioritize ongoing monitoring and evaluation to assess effectiveness in specific contexts.

### 5.2. Implications for Research

Based on the findings from this systematic review, the evidence regarding the effects of interventions aimed at improving vaccination uptake among adults is insufficient, as only 35 studies were included in our review. All these studies were conducted in high-income countries (HICs), which suggests that the results may not be applicable to low- and middle-income countries (LMICs). To better understand the effects of these interventions and to apply them across various settings, research needs to be conducted in LMICs as well. Furthermore, future studies in both HICs and LMICs should investigate and report on the cost-effectiveness of implementing interventions to increase vaccination uptake among adults. This review also identified low-certainty evidence regarding the effectiveness of recipient-oriented and provider-oriented interventions, highlighting the need for future research to expand on other strategies that could enhance vaccination uptake among adults. This review has limitations, specifically the exclusion of cluster RCTs, which may have narrowed the scope of evidence. By excluding cluster RCTs, we may have missed significant data that could impact the generalizability of the findings. The fact that most of these studies were conducted in high-income countries further complicates the ability to generalize the results to low- and middle-income contexts.

## Data Availability

The raw data supporting the conclusions of this article will be made available by the authors on request.

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
