# Peer review of "Interventions to Improve Vaccination Uptake Among Adults: A Systematic Review and Meta-Analysis"

_vaccines, 2025, doi:10.3390/vaccines13080811_

Round 1

Reviewer 1 Report

Comments and Suggestions for Authors

The authors analyzed interventions to improve vaccination uptake among adults. The meta-analysis yielded interesting results that are systematically presented in the manuscript as well as conclusions that may help authorities to design appropriate strategies. However, I have some substantive comments that may help improve the quality of the manuscript and therefore I suggest authors to revise the manuscript in accordance with the suggestions in the attached document.

Comments and Suggestions for Authors

The authors analyzed interventions to improve vaccination uptake among adults. The meta-analysis yielded interesting results that are systematically presented in the manuscript as well as conclusions that may help authorities to design appropriate strategies. However, I have some substantive comments that may help improve the quality of the manuscript. Comments are listed below:

Abstract

General comment:

  • Please add the total sample size ( how many participants overall were included in the meta-analysis?)
  • Highlight if there were any significant limitations of the study that could affect the generalizability of the findings.

Introduction

General comment: The introduction is too extensive. Section 1.1 contains a Table 1 with data taken from various references showing recommended vaccines for adults and healthcare workers from 2018 onwards. Apart from the technical shortcomings of the Table 1 which make it extremely confusing for readers, I believe that the data presented is not up to date and that it would be better to present data from WHO, ECDC and other databases and references providing more recent data on recommended vaccines for adults. Otherwise, a single paragraph is sufficient, listing only references that would provide the reader with an up-to-date overview of recommended vaccines for adults. Besides, it is necessary to shorten section 1.2 and merge sections 1.3-1.5 into one. I also suggest avoiding repetitions (e.g. check the sentences in lines 150-151 , lines 167-168 and  in lines 181-182).

In section 1.5, authors list older people as a risk group for vaccine-preventable diseases. However other target groups of adults for vaccination should also be listed (e.g. people with comorbidities regardless of age, pregnant women, etc.).

Methods

General comment: This section (Methods) is also too extensive. I would suggest that the methodology be presented briefly, divided into a maximum of three subsections, one of which is a statistical analysis, and that the detailed methodology be presented in a supplement.

I checked but was unable to find in the Methods the information shown in the Abstract about the time the search was done ( (August 2021 and November 2024). I was also unable to find what timeframe was set for randomized trials.

Also the age of participants (≥ 18 years) as inclusion criteria should be mentioned in Methods.

Results

General comment: PRISMA flow diagram could be helpful to present the flow of information through the different phases of a systematic review and make the display of results more understandable. At the same time, by introducing this flow diagram as a supplement, the text in section 3.1 would be significantly shortened.

Consider moving this section (3.1) to the Methods section. In that way the Results would start from the section 3.2. ( Effects of interventions).

Discussion

  • To avoid repetition, I advise deleting the text on lines 589-594.
  • Clearly specify the limitations of the study. Text in the lines 710-714 could be serve as a basis.

Author Response

Editor and reviewers' comments have all been addressed and included in the attached cover letter. In the manuscript, they have been highlighted in red.

Reviewer 2 Report

Comments and Suggestions for Authors

The paper is interesting and well written. The authors investigated the effects of interventions aimed at increasing vaccination (smallpox, poliovirus) uptake among adults. The review confirmed that letter reminders and education directed at recipients are effective in increasing vaccination uptake compared to the controls.  I suggest to consider for discussion and add as references the 2 positions on smallpox by Cabanillas et al. published in allergy.

Author Response

(The authors gave the same response as above.)
